# Comparing Changes in IPV Risk by Age Group over Time in Conflict-Affected Northeast Nigeria

**DOI:** 10.3390/ijerph20031878

**Published:** 2023-01-19

**Authors:** Bolatito O. Ogunbiyi, Beth J. Maclin, Jeffrey B. Bingenheimer, Amita Vyas

**Affiliations:** Department of Prevention and Community Health, Milken Institute School of Public Health, George Washington University, Washington, DC 20052, USA

**Keywords:** intimate partner violence, adolescent girls and young women, Nigeria, Boko Haram, conflict setting

## Abstract

Increased risk of intimate partner violence (IPV) has been well documented among women and girls living in conflict zones. However, how residence in a conflict area differentially impacts adolescent girls and young women (AGYW) compared to older women is less understood. This paper examines whether the levels of IPV changed more among AGYW compared to older women in six Boko Haram (BH)-affected States in Nigeria. The Nigeria Demographic and Health Survey data was used to compare the level of the three types of IPV (emotional, physical, and sexual) among AGYW compared to older women before and during the BH conflict (2008 and 2018). We ran a multiple linear regression model with an interaction term for ever-partnered female respondents living in six Northeast States, adjusting for relevant covariates. A significantly higher proportion of both older and younger women reported experiencing emotional and sexual IPV in 2018 than in 2008, with a higher increase reported among AGYW. Sexual IPV increased by six percentage points more among AGYW compared to older women. AGYW in the BH-affected States are more vulnerable to experiencing sexual IPV relative to older women. This study highlights the need for youth-focused IPV interventions in the BH-affected States.

## 1. Introduction

Intimate partner violence (IPV) which includes physical, sexual, and emotional violence by an intimate partner remains pervasive and is the most prevalent form of violence against women and girls globally, affecting up to 641 million [1]. At least one in three women are subjected to physical or sexual violence by an intimate partner in their lifetime [1,2]. These numbers are likely to be significantly higher given the high levels of the associated stigma and underreporting of sexual abuse among women and girls [1]. According to a comparative analysis of the Demographic and Health Survey (DHS) data from ten countries (Bangladesh, Bolivia, the Dominican Republic, Haiti, Kenya, Malawi, Moldova, Rwanda, Zambia, and Zimbabwe), the prevalence of physical and sexual IPV perpetrated by a current or most recent husband or cohabiting partner of ever-partnered women ranged from 16% in the Dominican Republic to 75% in Bangladesh [3]. In the same study, the prevalence of sexual IPV ranged from 3% in Moldova to 26% in Bangladesh [3].

IPV affects young women and girls as much as older women, with younger women sometimes at a higher risk than older women [4,5,6]. Whether they are in formal unions in contexts where early marriage is pervasive or within informal relationships such as dating, young women and girls experience IPV. Younger women are at the highest risk for recent violence, reporting the highest rates of past year IPV at 16%, and by their mid-twenties, one in four young women aged 15–24 years who have been in a relationship would have already experienced IPV [1]. There is mounting evidence that young age is a risk factor for women’s experience of IPV in different contexts, including conflict-affected rural Uganda, Canada, and the U.S. [7,8,9,10]. Even after controlling for other important factors such as the partner’s alcohol use, younger women in these population-based studies were more likely to have experienced IPV in the last 12 months compared to their older counterparts [7,9,10]. For example, young women are more at risk of rape than older women [11]. Some studies suggest that early marriage or cohabiting, particularly being married before the age of 15, may increase the risk of experiencing IPV [12,13,14]. These findings have been attributed to increased power differentials when younger girls often from poorer households and with low educational attainment marry older men with more education and better livelihood outcome, resulting in less power leverage by the girls in their relationships [15]. This evidence suggests that young women, particularly in settings where IPV is common, are most at risk of experiencing IPV compared to older women.

A growing body of evidence also suggests that living in a conflict-affected setting is associated with increased levels of IPV experienced by women as social protection systems deteriorate, exacerbating the conditions that predispose women to risks for violence [16,17]. Indeed, IPV remains the most common form of gender-based violence (GBV) experienced by women and girls in these settings [18]. Three studies on IPV among women in conflict-affected northern Uganda reported prevalence rates ranging from 53 to 55 percent [7,19,20]. A population-based survey found elevated rates of ongoing emotional and physical IPV among internally displaced populations in Uganda [21]. In a study that examined the levels of IPV in Liberia, residence in a conflict-affected district was associated with a 50 percent increase in IPV risk [17]. Another study that examined the correlates of IPV among urban women in 27 sub-Saharan African countries reported significantly higher IPV levels in conflict and post-conflict states [12]. A study in Nigeria found that residence in armed conflict-affected areas increased the probability of experiencing physical or sexual IPV by 3.7 percentage points and was associated with lower progress in the reduction of physical and sexual IPV [13].

While the literature on women’s experiences of IPV in conflict-affected settings is still relatively nascent, there is limited evidence on the levels of IPV among conflict-affected adolescent girls and young women (AGYW) aged 15–24 years, particularly how their experiences differ from older women’s. The limited data that does exist suggests that adolescent girls and young women are particularly vulnerable to IPV. Murphy and colleagues found that South Sudanese girls who were directly exposed to the armed conflict were more than seven times more likely to experience non-partner sexual violence and more than twice likely to experience IPV compared to those who were not directly exposed to armed conflict [22]. Another study found that almost 86% of displaced AGYW living in informal settlements in Uganda had experienced IPV in the last 12 months [23]. Among conflict-affected communities in the Democratic Republic of Congo (DRC) and Ethiopia, more than half of the girls (51.62%) had experienced at least one form of violence in the previous 12 months [24].

In 2009, Boko Haram, the Sunni Islamist non-state armed group in Nigeria, escalated from largely being non-violent to a more regimented and tactical insurgency [25]. More than 15,000 civilians died between 2011–2018 due to fighting between Boko Haram and the Nigerian military; millions have also been displaced. All six states in the Northeast region of Nigeria have experienced considerable casualties from Boko Haram attacks, particularly since the escalation of fighting in 2013 [26]. For this study, we focused on the six states—Adamawa, Bauchi, Borno, Gombe, Taraba, and Yobe- in the Northeast region as they have all been heavily affected by the Boko Haram conflict.

GBV is a defining feature of the Boko Haram conflict. Boko Haram gained international notoriety after attacking a school in Chibok—a town in Borno state in 2014, where Boko Haram is most active—and kidnapping more than 200 schoolgirls and forcing them into sexual slavery [27]. The group’s wider repertoire of violence is marked by a focus on violence against women and girls, including forced labor, mental and physical abuse, forced religious conversion, abduction, forced marriage, and rape; the last two are not only damaging in the present, but also pose long-term health consequences related to pregnancy and sexually transmitted infections [28].

According to the Nigeria DHS conducted in 2008 and 2018, a period that corresponds to before and after the onset of the Boko Haram insurgency, the national IPV rate (emotional, physical, or sexual IPV in the last 12 months) declined from 30.8% to 29.5% [29,30]; however, the IPV rate in the Northeast region increased from 27% in 2008 to 41.3% in 2018 (based on authors’ calculation). Given the numerous Boko Haram-led attacks in the Northeast region within this period [31], the rise in IPV could be conflict-related.

While the rise in IPV prevalence in the Northeast region is evident, it is not clear if the trend is similar among AGYW compared to older women. In this paper, we examined whether the levels of IPV differentially changed among AGYW compared to older women in Nigeria’s conflict-affected Northeast region. We hypothesized that the conflict would have a larger impact on the prevalence of IPV among adolescent girls and young women compared to older women.

## 2. Materials and Methods

### 2.1. Data Sources and Study Population

We used the 2008 and 2018 Nigeria DHS women’s datasets, specifically the subset of respondents living in the Northeast region. The DHS program collects nationally representative household data from 15–49-year-old residents (male and female) in developing countries and countries that receive development support from the U.S. government. The 2008 Nigeria DHS was collected in June-October of that year, and the 2018 Nigeria DHS was collected in August-December of that year. The former year used the DHS-V instrument, while the latter used the DHS-VII instrument; both included the domestic violence module [29,30].

A total of 5436 AGYW (15–24 years) participated in the surveys in both years (2361 in 2008; 3075 in 2018). For older women (25–49 years), 8420 individuals participated in the surveys both years (3856 in 2008; 4564 in 2018). The DHS survey refers to the section on IPV as the domestic violence (DV) module. When referring to the module, we use this language. However, we limited the scope of this paper to IPV, not all forms of DV. In the two survey years, 2223 AGYW and 3715 older women participants were selected for and completed the DV module. In 2008, 1592 AGYW were selected for and completed the DV module; 1235 of these respondents were ever-partnered. Among the older women, in 2008, 2624 respondents were selected for and completed the module; 2583 of these women were ever-partnered. In 2018, 631 AGYW were selected for and completed the DV module; of those, 432 respondents were ever-partnered. Lastly, 1091 older women were selected for and completed the module in 2018; of those, 1063 women were ever-partnered. The analytic sample for this paper was obtained from the ever-partnered respondents who completed the DV module.

Different sampling strategies were used to identify female respondents for the DV module which likely explains the smaller percentage of respondents who completed the module in 2018. In 2008, the DV module was administered to one eligible woman randomly selected in each household [29], while in 2018, the DV module was administered in the subsample of households that were selected for the men’s survey [30]. Further, in 2018, one percent of the selected respondents could not complete the module because of lack of privacy; this accounted for nine women in the Northeast region. With the added special weights, however, the 2018 sample can be considered nationally representative.

### 2.2. Measures

#### 2.2.1. Dependent Variables

The DHS program collects data on IPV experienced by female respondents currently or previously in marital partnerships or cohabiting with a man. Emotional, physical, or sexual violence perpetrated by other known or unknown perpetrators was not included in this analysis. There are three dependent variables of interest, one for each type of reported IPV on which data was collected: emotional, physical, and sexual. Ever-partnered respondents who completed the DV module were asked a series of three questions regarding experiences of emotional violence, seven regarding physical violence, and two regarding sexual violence from a male spouse or partner. If they answered yes, they were then asked if it occurred often, sometimes, or not at all in the past 12 months. Respondents who said they experienced any of the specific instances of violence in the past 12 months received a score of one, while those who said they did not experience any of them in the past 12 months received a zero. This process was repeated for each type of IPV. These questions are standard measures, and the separation by IPV type has been employed elsewhere [19,22,24,32,33].

#### 2.2.2. Independent Variables

There were two independent variables of interest for this analysis. The first is time. Data collected in 2008 (coded as 0) is the pre-conflict reference as Boko Haram was not yet active during the survey reference period in 2008. Data collected in 2018 (coded as 1) is the conflict point as Boko Haram has been active for almost a decade at this time. The 2013 Nigeria DHS data was not used here because an analysis comparing IPV rates in 2008 and 2013 already exists [13], and data from 2018 allows us to understand the longer-term impacts of the ongoing Boko Haram conflict. The second independent variable is the age group: 15–24-year-old adolescent girls and young women (coded as 1) and 25–49-year-old older women (coded as 0). This distinction is based on the United Nations’ definition of youth [34].

### 2.3. Covariates

We considered the following established drivers of IPV in low- and middle-income countries as covariates: age, education, religion, ethnicity, employment, wealth, marital status, number of children, residence (rural/urban), state of residence, being married as a child, acceptance of wife-beating, their father beating their mother, partner’s age, partner’s education, partner’s employment, and alcohol use by partner [19,22,24]. Of particular interest given the conflict focus for this paper is that a male partner’s alcohol consumption increases a woman’s risk of IPV in a conflict-affected setting in northern Uganda [35] and Nigeria [13], and in post-conflict settings in eastern DRC [36]. Women and adolescent girls in conflict-affected areas who witnessed their fathers beating their mothers [13,35] and those who agree with more gender-inequitable attitudes such as wife beating [13,22] are also more likely to report having experienced IPV.

### 2.4. Analysis

To examine whether the respective age groups of respondents are comparable on each of the relevant demographic variables at each time point, we ran two-sample t-tests on continuous variables and chi-square statistical tests on categorical variables. We adjusted for variables that differed for either or both age groups between time points in the multiple linear regression model.

Further, we compared the proportion of each age group who reported experiencing each type of IPV and the proportion of younger and older respondents who were partnered (currently married or living with a partner; or divorced, widowed, or separated) between the two-time points using a two-sample test of proportions. These comparisons address two underlying conditions that might be at play. The first is the likelihood of experiencing IPV given that a respondent is currently or previously partnered. The second is the likelihood of being partnered, which may increase over time in conflict-affected settings given that families tend to arrange marriages for their daughters during the protracted conflict as a means of protecting the girls from the conflict or for economic gains associated with such marriages [27,28]. The latter comparison helps us to understand whether a change in the reported IPV within the two groups (AGYW and older women) stems from more ever-partnered women and girls experiencing IPV or more women and girls getting married or cohabiting with intimate partners over time.

We ran a multiple linear regression (MLR) model with an interaction term to examine whether the levels of IPV changed more among AGYW than among older women between 2008 (before the onset of the Boko Haram conflict) and 2018 (during the Boko Haram conflict). We ran the following model to test the impact of age on adolescent girls’ and women’s experiences of each type of IPV:IPV_ily_ = β0 + β1Year_y_ + β2AgeGroup_l_ + β3Year_y_*Agegroup_l_ + β4X_ily_ + ε_ily_.(1)

The outcome, IPV_ily_, is whether respondent ‘i’ reported experiencing any IPV (emotional, physical, and sexual, respectively), during year ‘y’, and while part of age group ‘l’. As earlier explained, the two independent variables, time (year ‘y’) and youth (age group ‘l’) are dichotomous variables, with one representing 2018 and being an AGYW, respectively, and zero representing 2008 and being an older woman, respectively. X_ily_ represents the included covariates (demographic variables that were significantly different across time points for each age group), and Ɛ_ily_ is the error term. The interaction term between year and age group, β3, is the key parameter of interest. We ran each of the three IPV-specific regression models for ever-partnered female respondents.

We ran all analyses in Stata 15 (StataCorp., Stata Statistical Software: Release 15. College Station, TX, USA) with a significance level of *p* < 0.05. Appropriate survey weights for the DV module as defined by the DHS sampling strategy were employed for all regression models.

## 3. Results

Compared to 2008, both age groups had completed more years of education than in 2018. Further, fewer participants in each age group reported being married before age 18 and an acceptance of a man beating his female partner (Table 1). In contrast, significant increases were found for both age groups reporting that their father beat their mother. Differences were also found for both age groups on religion and ethnicity between the two years. For the AGYW, fewer reported being married or living with a partner and living in an urban location in 2018 compared to 2008. For the older women, fewer were in the lowest wealth index categories in 2018 compared to 2008, and participants reported fewer children ever born. Both age groups had increases in their partners’ educational attainment and alcohol consumption, and reductions in their partners’ employment status between 2008 and 2018.

Among ever-partnered respondents, a significantly higher proportion of respondents from each age group reported experiencing emotional and sexual IPV in 2018 than in 2008 (Table 2). The differences between the two time points are larger for AGYW- emotional IPV increased by 21.75% (*p* < 0.001) among AGYW compared to the increase of 19.66% (*p* < 0.001) among older women, while sexual IPV increased by 10.26% (*p* < 0.001) among AGYW compared to the increase of 3.58% (*p* < 0.001) among older women. A significantly higher proportion of ever-partnered AGYW participants also reported experiencing physical IPV in 2018 compared to 2008 (7.53% increase (*p* < 0.001).

Being part of the younger age group increased the probability of an ever-partnered female respondent reporting sexual IPV by six percentage points, adjusting for covariates (*p* < 0.05, 95% CI (0.01, 0.10)) (Table 3). For emotional and physical IPV, the interaction terms were not significant. However, the probability of all ever-partnered respondents experiencing emotional IPV increased by 19 percentage points between 2008 and 2018 (*p* < 0.001, regardless of age group.

For both age groups, fewer respondents are married or living with a partner in 2018 compared to 2008, and more are divorced, separated, or widowed, or never partnered (Table 4). Excluding younger women being divorced, widowed, or separated, all the changes between 2008 and 2018 were statistically significant.

## 4. Discussion

Our findings add to the limited body of evidence on the vulnerability of adolescent girls and young women to IPV in a conflict-affected setting relative to older women. Overall, the findings confirm the hypothesis that residence in the Boko Haram conflict-affected area has a larger impact on the prevalence of IPV (sexual) among adolescent girls and young women compared to older women. Our findings also confirm an increase over time in the probability of experiencing sexual and emotional IPV among older and younger women, in line with the current literature on IPV in conflict settings [7,8,9,10,11]. Regardless of the age group, the two-to-three-fold increase in sexual and emotional IPV among women and girls in the Northeast region found in this study is troubling when considered from the angle of reducing national IPV rates [30]. It is worth noting that the increase in IPV rates reported in this study differs among the two groups of women, especially for sexual IPV—the probability of experiencing sexual IPV is six percentage points higher among adolescent girls and young women relative to older women after controlling for relevant IPV correlates in the Boko Haram -affected Northeast region of Nigeria. Given that sexual violence is usually underreported due to barriers such as fear of retribution or ridicule, stigma, and norms of secrecy, noting a significant change in reported sexual IPV among AGYW compared to older women is particularly alarming given that the numbers are likely underestimated [1,37].

The finding that fewer girls were ever-partnered as the Boko Haram conflict protracted deviates from past research which suggests that more girls become partnered in conflict settings as more girls are likely to be out of schools due to the breakdown of the educational system and other social protection systems, and families tend to marry their daughters to protect them from the conflict [22,27,28,38]. We note that the departure of this finding from past research is unlikely due to the gains made in girls’ education and addressing early marriage in the country given that similar gains have not been registered in the Northeast region since the beginning of the insurgency [39,40]. Rather, during this period some schools were closed for more than three years while others served as shelters to internally displaced people and as military bases, further limiting girls’ access to safe learning spaces and increasing likelihood to be partnered as the conflict protracted [40].

This finding, however, is critical to the explanation of the findings on sexual IPV as it suggests that the reported higher levels of sexual IPV among the AGYW are not the outcome of girls and young women being in forced or early marriages or relationships where they are exposed to experiencing IPV but may rather imply an increase in the perpetration of IPV by the partners of these women and girls [16,17]. This finding could be explained by the notable controlling behavior of husbands/partners of women and girls in conflict-affected settings reported by several studies to be a key predictor of IPV [12,13,18,23,36]. We also know from past research that power differentials in relationships between younger girls and older men predispose these girls to experience IPV [10,11,22,23]. The significant increase in the level of physical IPV among AGYW (reported in the bivariate analysis) corroborates the increase in the sexual IPV levels among this age group as these two types of IPV (sexual and physical) are often consider the most severe forms of IPV [4].

Higher levels of emotional IPV experienced by ever-partnered women and girls reported in this study mirror the overall trend of IPV in the region between 2008 and 2018 with an increased likelihood of women and girls experiencing emotional IPV since the onset of the conflict. This finding confirms previous studies conducted in Jordan and Lebanon [33]. This finding could be explained by the notable controlling behavior of male partners in conflict-affected settings [7,22,36] and by the higher likelihood of women facing violent backlash from their husbands after taking on more economic responsibilities during conflict, a finding that has been reported among West African women [41].

Beyond the Boko Haram conflict in the study area, another factor that might explain the age group differences in the reported IPV levels (particularly sexual IPV) is the dramatic change in IPV reporting landscape within the timeframe covered in this study. Prior to the enactment of the Violence Against Persons Prohibition Bill in 2015, there was no federal law against domestic violence, and only a few states had such laws [42]. Social media has also increasingly become a tool for Nigerian women to challenge the norm of silence around IPV since 2015 with many women openly sharing their experiences, thereby emboldening other women to also speak out [43,44]. This change in the cultural landscape could have implications for IPV reporting behaviors. This shift might be greater for younger women, who tend to be more educated and exposed to social media, compared to older women. Taken together, the evolving security, legal, and social landscape in the study context might explain the higher probability of AGYW reporting sexual IPV compared to their older counterparts. Additional research is needed to understand how these other factors might, individually and collectively, explain any changes in both experiences and reporting of sexual IPV in Boko Haram-affected states.

### Limitations

This study has a few limitations. Due to the presence of Boko Haram and the deteriorating security situation in 2018, data collection did not occur as planned. One cluster of census enumeration areas as well as 11 of the 27 selected LGAs in Borno state were dropped and replaced with clusters from the other 16 accessible LGAs [30]). These exclusions and replacements could mean that the results from Borno, the most conflict-affected state in the Northeast region, are not representative of the state’s population. As Ekhator-Mobayode and colleagues noted in their analysis of the 2013 DHS data, this could suggest that the results reported here are an underestimate of the true probability of adolescent girls and women living in the context of the Boko Haram conflict experiencing IPV [13]. While the data collected from one of the six states in Northeast, Nigeria does affect its representativeness, it does not negate the results we found given the likelihood of underreporting IPV. Rather, the results presented here should be interpreted as underreported, particularly because of the limitations in data collection in the most conflict-affected state in the region.

Second, this analysis does not explicitly measure exposure to conflict, but rather assesses the risk of being younger and female in a conflict setting. This means the results cannot be understood as establishing a causal relationship between the Boko Haram conflict and IPV outcomes. More research utilizing Global Positioning System (GPS) coordinates and Armed Conflict Location and Event Data Project (ACLED) data, similar to the work of Ekhator-Mobayode and colleagues [13] and Kelly and colleagues [17] could help explain how the exposure to conflict informs the vulnerability of girls and young women. However, the results mirror much of the other conflict-focused research found suggesting that the Boko Haram conflict is, at least, likely a contributing factor.

Lastly, because the analysis is exclusively focused on Northeast Nigeria, the analysis might miss larger nationwide trends that could help explain the results beyond the conflict context. Additional research that either looks at these trends nationwide or through a comparison of the Northeast with a comparable non-conflict-affected region would further help explain how the Boko Haram context informs the results presented in this study.

## 5. Conclusions

This paper revealed a higher level of sexual IPV among AGYW relative to older women in an armed conflict-affected area, indicating that being young is a factor that puts women at increased risk of IPV in an active conflict setting. Exposure to GBV such as IPV during adolescence and young adulthood, important periods that establish health and well-being trajectory, can compromise women and girls’ physical, psychological, and economic well-being in the future [45,46]. This finding indicates the need for interventions to address the consequences of IPV among AGYW, including unplanned pregnancies, sexually transmitted infections (STIs), abortion, and mental health issues [47]. These interventions, which should ideally be multisectoral involving the health, justice, security, and shelter systems, and should be tailored to ever-partnered AGYW in the Boko Haram-affected Northeast region as their needs and potential barriers to accessing care may be different from that of older ever-partnered women. These programs should seek to address the attitudes, behaviors, and social norms that perpetuate IPV at the individual and community levels in the conflict-affected region [48,49] given the emerging evidence on the effectiveness of such interventions in reducing violence against IPV among young women and girls in conflict and humanitarian settings.

## Figures and Tables

**Table 1 ijerph-20-01878-t001:** Demographics by Age Group and Data Collection Year of Ever-partnered Female Respondents who Completed the Domestic Violence Module in Northeast, Nigeria (mean (SD) or frequency (% within the group).

Characteristics	Adolescent Girls and Young Women (15–24)		Older Women (25–49)	
	2008 (*n* = 1235)	2018 (*n* = 432)	2008 (*n* = 2583)	2018 (*n* = 1063)
Age (*/ )	19.80 (2.53)	20.09 (2.33)	34.12 (7.20)	33.91 (6.87)
Years of Education (***/***) ^a^	1.77 (3.49)	2.46 (4.18)	2.07 (3.93)	3.33 (5.09)
Marital Status (*/ )				
Married or living with a partner	1209 (97.89%)	414 (95.83%)	2462 (95.32%)	1005 (94.54%)
Widowed, Divorced, or Separated	26 (2.11%)	18 (4.17%)	121 (4.68%)	58 (5.46%)
Location (*/ )				
Urban	253 (20.49%)	66 (15.28%)	591 (22.88%)	264 (24.84%)
Rural	982 (79.51%)	366 (84.72%)	1992 (77.12%)	799 (75.16%)
Wealth Index ( /***)				
Poorest	636 (51.50%)	206 (47.69%)	1330 (51.49%)	407 (38.29%)
Poorer	299 (24.21%)	123 (28.47%)	563 (21.80%)	272 (25.59%)
Middle	175 (14.17%)	62 (14.35%)	388 (15.02%)	213 (20.04%)
Richer	106 (8.58%)	33 (7.64%)	237 (9.18%)	116 (10.91%)
Richest	19 (1.54%)	8 (1.85%)	65 (2.52%)	55 (5.17%)
Religion (*/***)				
Islam	1077 (87.21%)	385 (89.12%)	1948 (75.42%)	822 (77.33%)
Christian or Catholic	140 11.34%)	47 (10.88%)	588 (22.76%)	240 (22.58%)
Other, Traditionalist, Missing	18 (1.46%)	0 (0%)	47 (1.82%)	1 (0.09%)
Ethnicity (*/**)				
Fulani	382 (30.93%)	139 (32.18%)	632 (24.47%)	240 (22.58%))
Hausa	257 (20.81%)	94 (21.76%)	400 (15.49%)	199 (18.72%)
Kanuri/Beriberi	214 (17.33%)	47 (10.88%)	351 (13.59%)	111 (10.44%)
Other/Don’t Know	382 (30.93%)	152 (35.19%)	1200 (46.46%)	513 (48.26%)
Number of children ( /***)	1.59 (1.35)	1.61 (1.21)	5.57 (2.95)	5.18 (2.70)
Married as a Child (**/***)				
Yes	1052 (85.18%)	364 (84.26%)	1875 (72.5%)	655 (61.62%)
No	183 (14.82%)	68 (15.74%)	708 (27.41%)	408 (38.38%)
Acceptance of Wife-beating (***/***) *	(*n* = 1221)	(*n* = 430)	(*n* = 2552)	(*n* = 1062)
Yes	702 (57.49%)	211 (49.07%)	1326 (51.96%)	461 (43.41%)
No	519 (42.51%)	219 (50.93%)	1229 (48.16%)	601 (56.59%)
Father Beat her Mother (***/***)	(*n* = 1225)		(*n* = 2570)	
Yes	60 (4.90%)	45 (10.42%)	231 (8.99%)	149 (14.02%)
No	1104 (90.12%)	361 (83.56%)	2189 (85.18%)	861 (81.00%)
Don’t know	61 (4.98%)	26 (6.02%)	150 (5.84%)	53 (4.99%)
Partner’s Age	(*n* = 1194)	(*n* = 414)	(*n* = 2427)	(*n* = 1005)
	29.56 (6.61)	30.09 (6.68)	44.58 (10.52)	44.25 (10.04)
Partner’s Education Attainment (***/***)	(*n* = 1223)	(*n* = 414)	(*n* = 2567)	(*n* = 1005)
None	782 (63.94%)	236 (57.00%)	1645 (64.08%)	503 (50.05%)
Incomplete primary	63 (5.15%)	9 (2.17%)	105 (4.09%)	27 (2.69%)
Complete primary	121 (9.89%)	29 (7.00%)	256 (9.97%)	88 (8.76%)
Incomplete secondary	55 (4.50%)	21 (5.07%)	135 (5.26%)	43 (4.28%)
Complete secondary	147 (12.02%)	71 (17.15%)	221 (8.61%)	190 (18.91%)
Higher	49 (4.01%)	41 (9.90%)	194 (7.56%)	141 (14.03%)
Don’t know	6 (0.49%)	7 (1.69%)	11 (0.43%)	13 (1.29%)
Partner’s Employment (***/***)	(*n* = 1223)	(*n* = 414)	(*n* = 2549)	(*n* = 1005)
Any	1219 (99.67%)	403 (97.34%)	2531 (99.29%)	971 (96.62%)
None	4 (0.33%)	11 (2.66%)	18 (0.71%)	34 (3.38%)
Partner’s Alcohol Use (*/**)	(*n* = 1234)	(*n* = 432)	(*n* = 2578)	(*n* = 1063)
Yes	43 (3.48%)	26 (6.02%)	261 (10.12%)	147 (13.83%)
No	1191 (96.52%)	406 (93.98%)	2317 (89.88%)	916 (86.17%)

* = *p* < 0.05; ** = *p* < 0.01; *** = *p* < 0.001. Note: Table 1 presents a comparison of the demographics of the two age groups at both time points. Only the statistically significant differences are presented (With the exception of the partner’s age); the non-significant demographic details are available upon request. ^a^ The value to the left of the forward slash represents the *p*-value of the comparison between AGYW participants at each time point. The value to the right of the forward slash represents the *p*-value of the comparison between older women participants at each time point.

**Table 2 ijerph-20-01878-t002:** Changes in Proportions of Reporting IPV in the Past 12 Months among Ever-partnered Female Respondents who Completed the Domestic Violence Module by Age Group (proportion (95% CI).

	AGYW			Older Women		
	2008	2018	% Point Change	2008	2018	% Point Change
Emotional	0.16 (0.14, 0.18)	0.38 (0.33, 0.42)	0.22 (0.17, 0.27) ***	0.22 (0.21, 0.24)	0.42 (0.39, 0.45)	0.20 (0.16, 0.23) ***
Physical	0.10 (0.08, 0.12)	0.18 (0.14, 0.21)	0.08 (0.04, 0.11) ***	0.16 (0.15, 0.17)	0.16 (0.14, 0.18)	−0.00 (−0.03, 0.03)
Sexual	0.05 (0.03, 0.06)	0.15 (0.11, 0.18)	0.10 (0.07, 0.14) ***	0.05 (0.04, 0.06)	0.09 (0.07, 0.11)	0.04 (0.02, 0.05) ***

*** = *p* < 0.001.

**Table 3 ijerph-20-01878-t003:** Multiple Linear Regression on Differential Changes in Trends of Reporting IPV in the Past 12 Months based on Age Group among Ever-partnered Female Respondents who Completed the Domestic Violence Module (𝛽, SE, *p*-value (95% CI) ^b^.

	Emotional IPV	Physical IPV	Sexual IPV
Time*Youth	−0.04, 0.03, 0.25 (−0.10, 0.03)	0.03, 0.02, 0.28 (−0.02, 0.07)	0.06, 0.02, 0.02 (0.01, 0.10)
Time	0.19, 0.02, <0.001 (0.14, 0.23)	0.01, 0.02, 0.56 (−0.02, 0.05)	0.03, 0.01, <0.01 (0.01, 0.05)
Youth	−0.01, 0.02, 0.46 (−0.05, 0.02)	−0.03, 0.02, 0.06 (−0.06, 0.00)	0.01, 0.01, 0.54 (−0.02, 0.03)
R-squared	0.0789	0.0930	0.0351

Note: ^b^ The MLR models were adjusted with the following covariates: respondent’s years of education, marital status, residence location, wealth index, religion, ethnicity, number of children, married before 18 years old, acceptance of wife beating, reporting their father beating their mother, partner’s employment, partner’s educational attainment, and partner’s alcohol use.

**Table 4 ijerph-20-01878-t004:** Changes in the proportion of all female respondents in Northeast Nigeria, split by age group and year, in relationships.

	Proportion (SE)	95% Con. Interval	% Point Change
Currently Married or Living with Partner			
AGYW (2008)	0.6518 (0.0098)	0.6324, 0.6708	−14.42 ***
AGYW (2018)	0.5076 (0.0090)	0.4900, 0.5253	
Older women (2008)	0.9357 (0.0039)	0.9275, 0.9430	−3.58 ***
Older women (2018)	0.8999 (0.0044)	0.8908, 0.9082	
Widowed, divorced, or separated			
AGYW (2008)	0.0224 (0.0030)	0.0172, 0.0293	0.36
AGYW (2018)	0.0260 (0.0029)	0.0209, 0.0323	
Older women (2008)	0.0454 (0.0034)	0.0594, 0.0035	1.40 **
Older women (2018)	0.0594 (0.0035)	0.0529, 0.0666	
Never partnered			
AGYW (2008)	0.3257 (0.0096)	0.3071, 0.3449	14.06 ***
AGYW (2018)	0.4663 (0.0009)	0.4488, 0.4840	
Older women (2008)	0.0189 (0.0022)	0.0151, 0.0238	2.19 ***
Older women (2018)	0.0408 (0.0029)	0.0354, 0.0469	

** = *p* < 0.01; *** = *p* < 0.001.

## Data Availability

The dataset used in this research are publicly available on the Demographic and Health Survey program website, https://dhsprogram.com/data/dataset/Nigeria_Standard-DHS_2008.cfm?flag=0 (accessed on 6 November 2019) and https://dhsprogram.com/data/dataset/Nigeria_Standard-DHS_2018.cfm?flag=0 (accessed on 6 November 2019).

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
