# Peer review of "Comparing Changes in IPV Risk by Age Group over Time in Conflict-Affected Northeast Nigeria"

_ijerph, 2023, doi:10.3390/ijerph20031878_

Round 1

Reviewer 1 Report

Dear Editors and authors,

I thank you for my consideration as a reviewer of this manuscript. It is my pleasure to contribute to International Journal of Environmental Research and Public Health.

This research contributes to the knowledge of intimate partner violence in relation to age and conflict-affected areas.

I would like to suggest the followings comments:

The introduction section is well-justified. I recommend indicating the areas included in the Demographic and Health Survey and the countries with the highest and lowest prevalence of IPV (lines 31-36). Moroever, I suggest writing the specific objective and hypotheses or research questions of this study.

In the methodology section, I suggest reorganizing the data analysis separately and, also, indicating the statistical analysis carried out.

In the discussion section, I would like to recommend reorganizing the structure. First, I suggest including the objetive in the first paragraph. Following, each paragraph would ask the hypothesis or research question with the results. Finally, including limitations and conclusions with recommendations for research and clinical areas. 

Author Response

Thank you for the feedback. Please see below our responses to the suggestions.

Point 1: I recommend indicating the areas included in the Demographic and Health Survey and the countries with the highest and lowest prevalence of IPV (lines 31-36).

Response: We have addressed this point, please to lines 33-35 of the revised manuscript

Point 2: Moreover, I suggest writing the specific objective and hypotheses or research questions of this study.

Response: The closing paragraph of the introduction section states the objective (lines 108-111); a clear hypothesis statement has been added (lines 111-113)

Point 3: In the methodology section, I suggest reorganizing the data analysis separately and, also, indicating the statistical analysis carried out. 

Response: The 'Analysis' section clearly presents how the data was inspected (chi-square statistical tests, two-sample t-tests, and two-sample test of proportions). Further, we do detail the statistical analyses undertaken (multiple linear regression with interaction term) with a clear presentation of the model equation. While we appreciate the reviewer's suggestions, we do not think it will improve the manuscript to separate this portion of the methods section into two sub-sections.

Point 4: In the discussion section, I would like to recommend reorganizing the structure. First, I suggest including the objective in the first paragraph. Following, each paragraph would ask the hypothesis or research question with the results.

Response: While we appreciate the reviewer's suggestions, we do not think it will improve the manuscript  to reorganize the ‘Discussion’ section by repeating the study objective. We have included a summary of our inference on the hypothesis based on the analyses in 279-281.

Point 5: Finally, including limitations and conclusions with recommendations for research and clinical areas. 

Response: We extensively discussed the key limitations of this study under the discussion (lines 341-368). For clarity, we have now included a ‘limitation’ subsection in line 340. We also presented the conclusions with recommendations for research under the ‘Conclusion’ section in lines 369-385.

Reviewer 2 Report

This is a well-written paper that analyzes longitudinally changes in IPV risk by age group among women and girls living in conflict zones. The methodology used by this study---utilizing a multiple linear regression model with an interaction term for ever-partnered female respondents living in 6 Northeast Nigerian conflict affected states among 2 different data sets from 2008 and 2018 from Nigerian Demographic and Health Survey data---is detailed and appropriate. The results of the study are of considerable significance to researchers exploring IPV risk for women in conflict affected areas. There are minor typographical issues in the text. Please be consistent in the use of acronyms. For instance, Boko Haram has been referred to both as Boko Haram and BH in the text. That can cause confusion. Also on page 3, 3rd paragraph under Materials and Methods please paraphrase the two quotes that are in there. It is not clear why they are being used as quotes.

Author Response

Thank you for the feedback. Please see below our responses to the suggestions.

Point 1: There are minor typographical issues in the text. Please be consistent in the use of acronyms. For instance, Boko Haram has been referred to both as Boko Haram and BH in the text. That can cause confusion.

Response: We have corrected the typographical errors in the document and we have removed the acronyms for Boko Haram (BH) in the body of the document. We have also maintained consistency in the use of other acronyms such as adolescent girls and young women (AGYW). 

Point 2: Also on page 3, 3rd paragraph under Materials and Methods please paraphrase the two quotes that are in there. It is not clear why they are being used as quotes.

Response: We have paraphrased the quotes texts and adjusted the in-text citations.

Round 2

Reviewer 1 Report

I want to thank the authors for their responses to comments and suggestions. If possible, I would like to recommend including citations in the hypothesis of this study. 

Author Response

We are grateful for the feedback. Please se below our response to the suggestion.

Point 1: I want to thank the authors for their responses to comments and suggestions. If possible, I would like to recommend including citations in the hypothesis of this study. 

Response: Respectfully, we prefer not to add citations to our hypothesis, as we developed the hypothesis ourselves rather than deriving them directly from another source. To be sure, our hypothesis is informed by numerous other sources, but those sources are already cited in the paper's Introduction. Our goal was to write an introduction that not only provides a compelling motivation for our hypotheses and, but also appropriately acknowledges how our thinking was shaped by others working in the field.
